# Functional comparison of human and murine equilibrative nucleobase transporter 1

**Chan S. Kim, Aaron L. Sayler, Hannah Dean, Nicholas M. Ruel, James R. Hammond** *

Department of Pharmacology, University of Alberta, Edmonton, Alberta, Canada

* james.hammond@ualberta.ca

## Abstract

6-Mercaptopurine (6-MP) maintenance therapy is the mainstay for various types of leukemia and inflammatory bowel disease. 6-MP is associated with numerous adverse effects including gastrointestinal intolerance, myelotoxicity, and hepatotoxicity. This can lead to therapy discontinuation which is associated with a higher risk of relapse. Drug transporter expression is a known factor contributing to patient variability in drug response and toxicity. We have established that the *SLC43A3*-encoded equilibrative nucleobase transporter 1 (ENBT1) mediates the transport of 6-MP into human lymphocytes and human embryonic kidney 293 (HEK293) cell lines transfected with *SLC43A3*. ENBT1 is known to be expressed in the gastrointestinal tract, bone marrow, and the liver. However, the relationship between ENBT1 and 6-MP-associated adverse events, and its pharmacokinetics, is unknown. To validate the use of mouse models (e.g. slc43a3-null mice) for exploring this relationship, we assessed the functional similarities between human and murine ENBT1 using HEK293 cells transfected with the respective *SLC43A3*/*slc43a3* constructs, and the leukemia cell lines MOLT-4 (human) and L1210 (murine). Based on *in silico* analyses of structural similarities between transporters, we hypothesized that human and murine ENBT1 will have similar 6-MP transport/inhibition kinetics and a similar impact on 6-MP-induced cytotoxicity. We show herein that *mslc43a3*-encoded mouse ENBT1 transports both [$^3$H]6-MP and [$^3$H]adenine with kinetics similar to those of *hSLC43A3*-encoded human ENBT1. Both are also similarly distributed in mouse and human tissues. Therefore, data obtained from mouse models where ENBT1 is disrupted or modified may provide clinically relevant insights on its roles in modulating the actions of 6-MP.

## Introduction

Thiopurines are a class of nucleobase analog drugs that play a pivotal role in cancer therapy due to their ability to disrupt DNA synthesis and suppress the immune system. Thiopurines operate as prodrugs, necessitating intracellular conversion into therapeutically active metabolites to exert their effects. One of the most established drugs in this category is 6-mercaptopurine (6-MP), which is utilized in the treatment of various cancers including acute lymphoblastic leukemia, and autoimmune conditions such as inflammatory bowel diseases

**Funding:** Canadian Institutes of Health Research, Operating Grant#168913. The funders had no role in study design, data collection and analysis, decision to publish, or preparation of the manuscript.

**Competing interests:** NO authors have competing interests.

[1]. While most patients tolerate 6-MP well, intra-individual variability of 6-MP bioavailability between patients continues to complicate the attainment of target therapeutic outcomes and the minimization of off-target toxicities [2, 3]. As a result, 6-MP therapy discontinuation due to low clinical efficacy or abundant adverse events is commonly reported, despite this increasing risk of relapse with lower survival rates [4, 5]. Observed variability is primarily attributed to genetic variations of enzymes involved in 6-MP metabolism [6–8]. However, the prodrug nature of 6-MP underscores the importance of not only metabolic pathways, but also drug transporters which precede intracellular metabolism. It is well established that variations in the expression of drug transporters have a significant impact on the therapeutic efficacy and adverse effects of numerous drugs [9], and this is likely to be the case for 6-MP as well. Several mechanisms have been proposed for the transport of 6-MP, including relatively low affinity uptake by nucleoside transporters encoded by the *SLC29* gene family [10, 11]. In 2007 we characterized a novel, relatively high affinity, nucleobase transport system in human cardiac microvascular endothelial cells and gave it the designation equilibrative nucleobase transporter 1 (ENBT1) [12]. Using transporter-selective inhibitors, we determined that ENBT1 was distinct from any known nucleoside/nucleobase transporters. However, the molecular identity of this nucleobase transporter remained unknown until *SLC43A3*/ENBT1 was characterized in Madin-Darby canine kidney cells as a transporter for endogenous purine nucleobases such as guanine, adenine, and hypoxanthine [13]. Our lab further characterized ENBT1 in stably transfected human embryonic kidney 293 (HEK293) cells, and showed that ENBT1 plays a major role in mediating the cellular accumulation of 6-MP, and sensitizing cells to the cytotoxic effects of 6-MP [14]. Nevertheless, how *SLC43A3*-encoded ENBT1 expression impacts 6-MP biodistribution and adverse effects remains unknown. To further investigate this relationship, it would be advantageous to employ transgenic mouse models. However, there is no literature to date on murine *slc43a3*-encoded mENBT1 and how it compares functionally to human *SLC43A3*-encoded hENBT1. Predicted transmembrane topology analyses [15] indicate that both display 12 transmembrane domains (TMD), a prominent extracellular loop between TMD 1–2, and a prominent intracellular loop between TMD 6–7 (Fig 1A). Pairwise sequence analysis [16] indicates that *SLC43A3*-encoded hENBT1–1476 bps, 491 aa (UniProt ID: Q8NBI5) and *slc43a3*-encoded mENBT1–1509 bps, 502 aa (UniProt ID: A2AVZ9) share 73.4% percent identity and 82.1% percent similarity in primary amino acid sequence (Fig 1B). One notable difference is the presence of a polyglutamine tract, in the aforementioned TMD 6–7 intracellular loop of mENBT1. Such differences in primary structure may impact transport function and substrate specificity. Therefore, elucidating the function of mENBT1 in mediating 6-MP transport and how it potentially differs from hENBT1 function is critical to assessing the clinical relevance of data obtained using mouse models. We report herein on the relative characteristics of adenine and 6-MP transport by mENBT1 and hENBT1, assessed in parallel, using both recombinant expression models and endogenously in leukemia cells lines.

## Materials and methods

### Materials

[$^{14}$C]-6-MP (50 Ci/mmol), [$^3$H]-6-MP (0.3 Ci/mmol), [2,8-$^3$H]-adenine (20 Ci/mmol), and [$^3$H]-water (1 mCi/g) were purchased from Moravek Biochemicals (Brea, CA). EcoLite liquid scintillation cocktail was purchased from MP Biomedical (Irvine, CA). Adenine, 6-MP, 6-thioguanine (6-TG), 3-(4,5-dimethylthiazol-2-yl)-2,5-diphenyltetrazolium bromide (MTT), dipyridamole, geneticin (G418 sulfate), Dulbecco's modified Eagle's medium (DMEM), fetal bovine serum (FBS), penicillin-streptomycin, and Immobilon®-P PVDF membrane were purchased from Sigma-Aldrich (St. Louis, MO). RPMI (Roswell Park Memorial Institute) -1640 medium

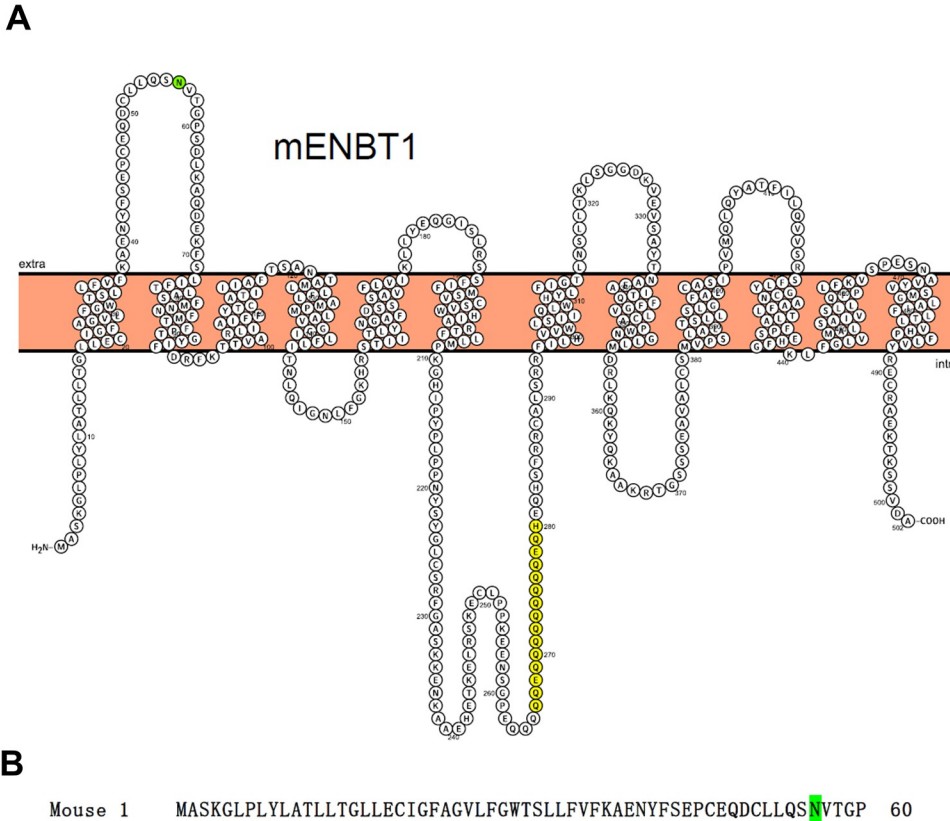

**Fig 1. Predicted membrane topology of mENBT1 and sequence homology with hENBT1.** (A) Membrane topology of mENBT1 was generated utilizing Protter version 1.0 [15]. The predicted N-glcosylation site at ASN56 is noted in green. The 15 amino acid glutamine/glutamate insert in mENBT1 is shown in yellow. (B) Pairwise sequence alignment of mouse and human ENBT1. The full amino acid sequence of mENBT1 is shown compared with hENBT1, with identical residues indicated with a dot and differing residues shown in red in hENBT1. The N-glycosylation site and poly glutamine/glutamate insert in the mouse ENBT1 are highlighted in green and yellow, respectively.

and ECL Prime western blotting system were purchased from Cytiva Life Sciences (Marlborough, MA). 6-Methylmercaptopurine (6-MeMP), oligo(dT)$_{12-18}$ primer, HEPES, bicinchoninic acid (BCA) kit, sodium pyruvate, 0.05% trypsin/EDTA, PowerTrack SYBR Green, HALT Protease Inhibitor Cocktail, ethylenediaminetetraacetic acid (EDTA), TRIzol Reagent, Taq DNA polymerase, SuperScript III Reverse Transcriptase, and the GeneArt Cloning pMA

plasmid were purchased from Thermo Fisher Scientific (Waltham, MA). 6-Thiouric acid (6-TU) sodium salt dihydrate were purchased from Toronto Research Chemicals (Toronto, ON). All primers were purchased from Integrated DNA Technologies (Coralville, IA). The HEK293, MOLT-4, and L1210 cells were purchased from ATCC (Manassas, VA). The primary antibodies were mouse monoclonal IgG1 anti-β-actin (C4, sc-47778, Lot #B0719; Santa Cruz Biotechnology Inc., Texas), mouse monoclonal IgG1 anti-MYC antibody (C-33, sc-42, Lot #3013479; Santa Cruz Biotechnology Inc., Texas), and rabbit polyclonal anti-hENBT1 (HPA030551, Lot #000001575; Sigma-Aldrich, St. Louis, MO). Secondary antibodies were anti-mouse IgG horseradish peroxidase (HRP)-linked antibody (#7076S, Lot #32; Cell Signaling Technology, Massachusetts), anti-rabbit HRP-linked antibody (#7074S, Lot #27; Cell Signaling Technology, Massachusetts), and mouse-IgGk BP-HRP (sc-516102, Lot #B0422; Santa Cruz Biotechnology Inc., Texas). C57BL/6 mice were bred in the Health Sciences Laboratory Animal Services unit at the University of Alberta. Mice were anesthetized with isoflurane and various tissues removed and placed immediately in TRIzol for subsequent processing for qPCR. Mice were then euthanized by exsanguination under anaesthesia. All animal work was conducted according to the standards of the Canadian Council on Animal Care using protocols (AUP00002022) approved by the Animal Care Committee of the Faculty of Medicine & Dentistry, University of Alberta.

### SLC43A3 plasmid construct and transfection

Oligonucleotides corresponding to the coding region of *SLC43A3* (NM_001278201) or *slc43a3* (NM_021398), with an N-terminal MYC-epitope tag, were prepared in the GeneArt Cloning pMA plasmid. The *MYC-SLC43A3* sequences were ligated into the mammalian cloning vector pcDNA3.1(-), after digestion with XbaI (5') and Kpn1 (3') restriction enzymes. The inserts were sequenced in both directions to confirm integrity, and then used to stably transfect wild-type HEK293 (HEK293-WT) cells, using the calcium phosphate method [17]. Cells expressing the transfected plasmid were selected based on their resistance to 600 μg/mL G418 sulfate, and expression of MYC-tag via dot blot. HEK293 cells stably transfected with the 'empty' pcDNA3.1(-) vector were also tested to assess the impact of the transfection procedure and chronic G418 sulfate treatment, on the cell line.

### Cell culture

Transfected-HEK293 cells were cultured in DMEM with 10% FBS, penicillin (100 U/mL), streptomycin (100 μg/mL), sodium pyruvate (1 mM), and G418 sulfate (300 μg/mL) to maintain selection pressure on the stable transfectants. Media was changed every 48 hours and cells harvested at approximately 80% confluence (typically 3–4 days after plating) by exposure to 0.05% trypsin/EDTA for 5–10 minutes at 37˚C in a humidified 5% $CO_2$ atmosphere. The resulting cell suspension was diluted with media containing serum and then centrifuged at 1000 x g for 5 minutes. L1210 cells were cultured in DMEM with 10% horse serum, penicillin (100 U/mL), and streptomycin (100 μg/mL). MOLT-4 cells were cultured in RPMI-1640 medium supplemented with D-glucose (4500 mg/L), 10% FBS, penicillin (100 U/mL), streptomycin (100 μg/mL), sodium pyruvate (1 mM), and HEPES (10 mM). L1210 and MOLT-4 cells were typically split 1:3 and incubated at 37˚C in a humidified 5% $CO_2$ atmosphere for 4 days prior to pelleting the suspended cells by centrifugation at 1000 x g for 5 minutes for use in the experimental procedures described below.

### Gene expression analysis

Cells from confluent 10 cm plates and tissues extracted from C57BL/6 mice of either sex were suspended in TRIzol reagent and homogenized for total RNA extraction according to the

**Table 1. Primers used for qualitative PCR and semi-quantitative PCR (qPCR) analyses.**

| Gene (Application) | Sequence | |
|---|---|---|
| *SLC43A3* (PCR) | Forward | 5'-GGAACTCCGCTCCTTCT-3' |
| | Reverse | 5'-GAAGTAGGACGTTCACTAGT-3' |
| *slc43a3* (PCR) | Forward | 5'-TTTACCCTGGCATCCTTCATG-3' |
| | Reverse | 5'-TAACGCGCATGAAAGGAAGA-3' |
| *slc43a3* (qPCR) | Forward | 5'-GAAACACCGTTCAACCATCATC-3' |
| | Reverse | 5'-CTGATGCCCTGCTCGTAAA-3' |
| *GAPDH* (PCR) | Forward | 5'-ACATCATCCCTGCCTCTAC-3' |
| | Reverse | 5'-TAAACCGATGTCGTTGTCC-3' |
| *Gapdh* (PCR) | Forward | 5'-GGGTGTGAACCACGAGAAATA-3' |
| | Reverse | 5'-AACAGAGGACGCTGAAGTTG-3' |
| *Gapdh* (qPCR) | Forward | 5'-GGGTGTGAACCACGAGAAATA-3' |
| | Reverse | 5'-GTCATGAGCCCTTCCACAAT-3' |

manufacturer's protocol. Total RNA concentration and purity were determined using a Nanodrop 2000 spectrophotometer (Life Technologies Inc). For qualitative polymerase chain reaction (PCR), 1 μg of total RNA was reverse transcribed to complementary DNA (cDNA) using Oligo(dT)$_{12-18}$ primer and Superscript III Reverse Transcriptase. Target cDNA sequences were amplified using recombinant Taq DNA Polymerase and primers designed for *SLC43A3*, *GAPDH*, *slc43a3*, and *gapdh* (Table 1). Primer efficiency and melt curves were assessed prior to their use for gene expression analysis. The following conditions were used for amplification: 3 minutes at 95°C, followed by 40 cycles of 30 seconds at 95°C; 30 seconds at 56°C; and 60 seconds at 72°C, followed by extension for 10 minutes at 72°C in a BioRad T-100 Thermocycler. Semi-quantitative PCR was conducted using cDNA prepared as described above with the primer sets shown in Table 1 using PowerTrack SYBR Green fluorescence on a Roche Light Cycler 480 System (Cardiovascular Research Centre, Edmonton, Canada). PCR conditions were: 2 minutes at 95°C, followed by 40 cycles of 15 seconds at 95°C, 60 seconds at 60°C, with a final melt curve analysis. Expression levels were calculated as $2^{-\Delta CT}$.

## Immunoblotting

Samples were extracted using RIPA buffer (150 mM NaCl, 50 mM Tris, 1% NP-40, 0.5% sodium dodecyl sulfate (SDS) containing HALT™ Protease Inhibitor Cocktail + EDTA. Extracted protein concentrations were assessed using a BCA assay kit and accordingly diluted to 1.5 μg/μL protein and adjusted to 2% 2-mercaptoethanol for reducing conditions. In some cases, 20 μg of protein from the cell lysates were incubated with 500 units of PNGase F at 37°C for 1 hr to de-glycosylate proteins. Proteins (20–30 μg) were resolved using sodium dodecyl sulfate-polyacrylamide gel electrophoresis (SDS-PAGE) on 12.5% acrylamide gels at 80 volts for 15 minutes, followed by 150 volts for 45 minutes, and transferred to Immobilon-P PVDF membranes at 25 volts for 30 minutes. Following the transfer, membranes were incubated with TBS-T (TBS-T; 150 mM NaCl, 50 mM Tris, pH 7.5, 1% Tween20) containing 3% bovine serum albumin (BSA), or5% skim milk powder for PNGase F treated membranes, at room temperature for 30 minutes (60 minutes for PNGase F treated membranes), to block nonspecific PVDF membrane binding. PVDF membranes were incubated for 16 hours at 4°C with anti-MYC, anti-hENBT1, or anti-β-actin primary antibodies diluted in TBS-T containing 5% BSA or skim milk powder, at 1:1000, 1:250, and 1:1000, respectively. Membranes were washed several times in TBS-T, then incubated for 1 hour at room temperature with the relevant HRP-

conjugated secondary antibody, anti-rabbit IgG-HRP or anti-mouse IgG-HRP diluted in TBS-T containing 5% BSA at 1:5000 or 1:3000, respectively. After further washing in TBS-T, membranes were treated with ECL Prime solution and visualized via chemiluminescence on an Amersham Imager 680 (GE Healthcare, Chicago, IL).

## Transport assay

Harvested cells were suspended in nominally sodium-free N-methyl-D-glucamine (NMG) buffer (140 mM NMG, 5 mM KCl, 4.2 mM $KHCO_3$, 0.36 mM $K_2HPO_4$, 0.44 mM $KH_2PO_4$, 0.5 mM $MgCl_2$, 1.3 mM $CaCl_2$, 10 mM HEPES, pH 7.4), to minimize potential contribution of sodium-dependent concentrative nucleoside transporter-mediated uptake. Additionally, cells were incubated for 15 minutes at room temperature with 10 μM dipyridamole (DY), an equilibrative nucleoside transporter (ENT) inhibitor, to block equilibrative nucleoside transporter–mediated uptake. The substrate uptake assay was initiated by adding 250 μL cell suspension to 250 μL [$^3$H] labeled substrate, layered over 21:4 silicone:mineral oil (200 μL) in 1.5 mL microcentrifuge tubes (conducted at room temperature). The uptake reaction was terminated after 2 seconds by centrifugation of the cells through the oil layer at ~10,000 $g$. This is the minimum incubation time achievable with this oil-stop technique, and it was found in previous studies that these short incubation times were necessary to assess the kinetics of this transporter [14, 18]. The aqueous layer was removed via vacuum-powered aspirator, and the tube was washed with 1 mL of NMG buffer prior to removal of the oil layer. The resulting cell pellet was digested in 250 μL 1 M NaOH overnight (~16 hours), with 220 μL aliquots of digested cells assessed for radioactive content using standard liquid scintillation counting techniques in a Beckman Coulter LS6500 scintillation system (Brea, CA).

In transfected-HEK293 cell lines, total uptake was defined as uptake of radiolabeled substrate in cells overexpressing recombinant hENBT1 or mENBT1. Non-mediated uptake was defined as the cell associated radiolabeled substrate in pcDNA3.1(-) empty vector transfected-HEK293 cells that are ENBT1-deficient. This non-mediated component would include passive diffusion into the cells and radiolabelled substrate remaining extracellularly in the digested cell pellet after processing the samples. In MOLT-4 and L1210 cell lines, total uptake was defined as the uptake of [$^3$H]6-MP in the absence of adenine, while non-mediated uptake was defined as the uptake of [$^3$H]6-MP in the presence of 5 mM adenine. ENBT1-mediated uptake was calculated as total uptake minus the non-mediated uptake.

To assess the ability of compounds to inhibit substrate uptake, cells were incubated with the inhibitor for 15 minutes prior to initiating the uptake assay, or exposed to the cells simultaneously with the radiolabelled substrate. In transfected-HEK293 cell lines, total uptake was defined as uptake of radiolabeled substrate in presence of an inhibitor (adenine or 6-MP); non-mediated uptake was defined as the uptake of radiolabeled substrate in the presence of 5 mM adenine; 100% uptake control was defined as uptake of radiolabeled substrate in absence of an inhibitor. After subtraction of the non-mediated uptake component, the degree of inhibition was calculated as the ratio of cellular accumulation in the presence of test compound to that measured in its absence (X 100).

Cell water volume (μL) was estimated by incubating cells with $^3H_2O$ for 3 minutes, centrifuging the cells through the oil layer at ~10,000 $g$, assessing 100 μL of the supernatant for radioactive content, and then processing as described above. Total cellular water volume was determined from the ratio of the decays per minute of the cell pellet, to the decays per minute of the supernatant, allowing for inter-experimental normalization via calculation of picomoles of substrate accumulated per microliter of cell-associated water.

## MTT cell viability assay

Transfected-HEK293, MOLT-4, and L1210 cells were seeded into 24-well plates at a density of $1.1 \times 10^5$ cells per well in complete culture medium. 1 hour following cell seeding onto plate, medium containing 6-MP, 6-MeMP, 6-TG, 6-TU (78 nM– 1.28 mM), or vehicle (DMSO) alone (Control) was added and incubated for 48 hours at 37˚C in a humidified 5% $CO_2$ atmosphere. Medium containing no cells was used to determine spontaneous MTT reduction (background). In transfected-HEK293 cells, media was removed via a vacuum-powered aspirator and replaced with 150 μL of Dulbecco's phosphate buffered saline (DPBS—137 mM NaCl, 2.7 mM KCl, 6.3 mM $Na_2HPO_4$, 1.5 mM $KH_2PO_4$, 0.5 mM $MgCl_2$, 0.9 mM $CaCl_2$, pH 7.4) containing MTT (1 mg/mL) for 90 minutes at 37˚C. In MOLT-4 and L1210, cell suspension was transferred to 1.5 mL microcentrifuge tubes and centrifuged for 10 minutes at 3000 rcf. Following centrifugation, media was removed via gentle pipetting and replaced with 150 μL of DPBS containing MTT (1 mg/mL) for 90 minutes at 37˚C. Microcentrifuge tubes were centrifuged again for 15 minutes at 23,500 rcf, and then DPBS containing MTT was removed via a vacuum-powered aspirator. The resultant formazan crystals were solubilized in 500 μL of DMSO and transferred to a 96-well plate in 200 μL duplicates. Absorbance was measured at 570 nM, in a SpectraMax® i3x Multi-Mode Detection Platform (Molecular Devices, San Jose, CA). After background subtraction, % cell viability was calculated as the ratio of absorbance in the presence of test compound to that measured with vehicle alone (X 100). In the case of a biphasic cell viability curve, only the first-phase $EC_{50}$ value was utilized in the analysis, as that component describes the population of cells that are actively proliferating and therefore most sensitive to antimetabolites such as 6-MP. While the second-phase likely describes the population of cells that are in a non-proliferative or quiescent state, and are therefore relatively insensitive to 6-MP after 48-hour treatment.

## Data analysis and statistics

Each assay was repeated 5–6 times using independent cell isolations and reagent preparations. In addition, within each assay, there were two or three technical replicates (using the same reagents and cells) that were averaged to obtain a single data point. Nonlinear curves were fitted to data (presented as mean ± SD from the 5–6 independent experiments), and statistical analyses were done using GraphPad Prism 10 software. In all cases, if the P-value determined from a statistical test was less than 0.05, the null hypothesis was rejected and the alternative hypothesis was favoured. Significant differences between groups were assessed using either Student's t-test or ordinary one- or two-way analysis of variance (ANOVA) followed by a post hoc Šidák multiple comparison test, as appropriate. For influx and cell viability analyses, significant differences in $K_m$ and $EC_{50}$ value, respectively, between the hENBT1 and mENBT1 data sets, were determined by the extra sum-of-squares F test. For inhibition analyses, the $K_i$ values were determined from the $IC_{50}$ values using the specified substrate concentration [S], based on the Cheng-Prusoff equation $K_i = IC_{50}/(1 + [S]/K_m)$ [19] and experimentally obtained $K_m$ values.

## Results

### Mouse tissue expression of *slc43a3*

Lung, heart, kidney, liver, spleen, duodenum, jejunum, and brain were chosen to represent tissues in humans that have been shown to express *SLC43A3* [20], and are relevant to 6-MP absorption and metabolism [1]. Lung tissue had the highest expression of *slc43a3*. Approximately 5-fold lower levels were observed in the heart, liver, kidney, and spleen. The intestinal

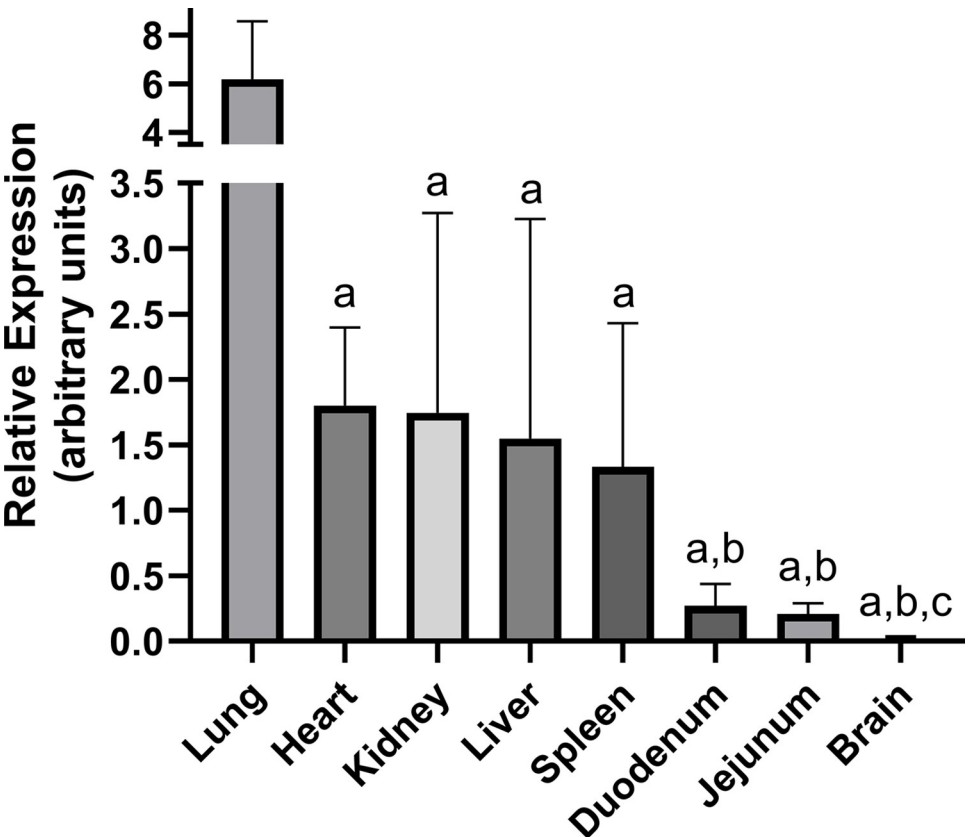

**Fig 2. Comparative expression of *slc43a3* in various mouse tissues.** Slc43a3 expression was determined by qPCR relative to *gapdh* (ΔCt) and shown in arbitrary units calculated as $2^{-\Delta CT}$x100. The primers used are shown in Table 1. Each bar represents the mean ± SD, n = 6. Statistical differences between ΔCt values were determined using one-way ANOVA (*a*–significantly different from lung; *b*–significantly different from *a*; *c*–significantly different from *a,b*).

tissues duodenum and jejunum had significantly lower levels yet (~7-fold lower than heart), and mouse brain showed negligible expression of *slc43a3* (Fig 2).

### *SLC43A3* transcript and ENBT1 protein expression

HEK293-WT cells that are innately ENBT1-deficient [12, 14] were stably transfected with *SLC43A3*, *slc43a3* or pcDNA3.1(-) empty vector, herein referred to as HEK293-hSLC43A3, HEK293-mslc43a3, and HEK293-EV, respectively. To confirm successful stable transfection of *slc43a3* and assess relative expression levels, qPCR and immunoblot were performed utilizing cDNA and cell lysates processed from HEK293-EV, HEK293-hSLC43A3, and HEK293-mslc43a3 cell lines (Fig 3). Assessment of PCR products confirm that HEK293-EV cells express minimal levels of *SLC43A3*, while the transfected cell lines show strong expression of their respective transcript. In terms of the leukemia cell lines tested, MOLT-4 cells express the *SLC43A3* transcript and L1210 cells express the *slc43a3* transcript, confirming primer species specificity (Fig 3). Immunoblotting showed that HEK293-hSLC43A3 cells express MYC-hENBT1 with a molecular mass of 56 kDa (similar mass was detected using either the human ENBT1 primary antibody or the anti-MYC antibody) (Fig 4A). HEK293-mslc43a3 cell membranes probed with the anti-MYC antibody revealed immunoreactivity with a molecular mass of approximately 60 kDa (Fig 4A). Unfortunately, despite testing a number of commercially available anti-mouse ENBT1 antibodies, none were found to be specific for mENBT1. Relative

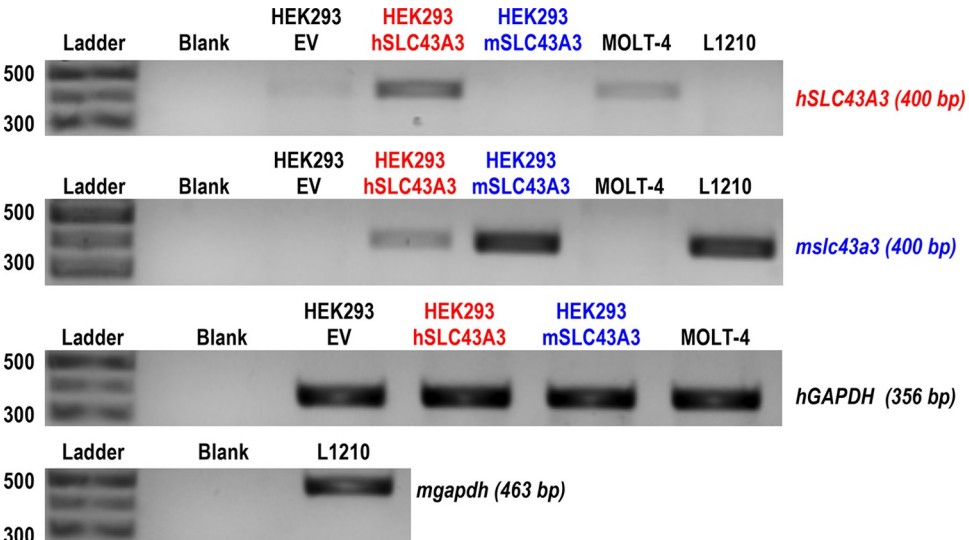

**Fig 3. Expression of *mslc43a3* and *hSLC43A3* in transfected HEK293 cells, MOLT-4 and L1210 cells.** Qualitative PCR was conducted using cDNA prepared from RNA extracted from cells transfected with pcDNA3.1 empty vector (HEK293-EV), vector constructs containing *hSLC43A3* (HEK293-hSLC43a3; shown in red) or *mslc43a3* (HEK293-mslc43a3; shown in blue), as well as from human MOLT-4 and murine L1210 leukemia cells. The lanes labeled 'Blank' are the control reactions using dH$_2$0 only. DNA ladders are shown with the corresponding base pair sizes for reference.

expression of mENBT1 and hENBT1 obtained by quantitative densitometry of the anti-MYC immunoblots is shown in Fig 4B, and indicate a similar level of protein expression in both cell lines. Treatment of membranes from the transfected cell lines with PGNase-F prior to immunoblotting resulted in a clear decrease in molecular mass of both hENBT1 and mENBT1, indicating that both proteins are natively glycosylated (Fig 4C).

## [³H]6-MP and [³H]adenine uptake in transfected HEK293 cells

HEK293-EV cells exhibited minimal uptake of [³H]adenine and [³H]6-MP (Fig 5A and 5C). However, HEK293 cells recombinantly expressing either mouse or human ENBT1 showed high levels of uptake of both [³H]adenine (HEK293-hSLC43A3, $K_m$: 93 ± 24 μM, $V_{max}$: 56 ± 6 pmol/μL/sec; HEK293-mslc43a3, $K_m$: 86 ± 22 μM, $V_{max}$: 62 ± 6 pmol/μL/sec) (Fig 5B) and [³H]6-MP (HEK293-hSLC43A3, $K_m$: 269 ± 67 μM, $V_{max}$: 121 ± 16 pmol/μL/sec; HEK293-mslc43a3, $K_m$: 253 ± 39 μM, $V_{max}$: 144 ± 12 pmol/μL/sec) (Fig 5D). The 6-MP and adenine associated $K_m$ and $V_{max}$ values were not significantly different between HEK293-hSLC43A3 and HEK293-mslc43a3 cells.

To investigate how adenine and 6-MP interact with ENBT1, inhibition assays were performed. In the case of a purely competitive inhibitor, the $K_m$ of a substrate should be greater than or equal to its $K_i$ as an inhibitor [19]. Adenine exhibited concentration-dependent inhibition of ENBT1-mediated [¹⁴C]6-MP uptake, with near complete inhibition at 1 mM adenine (Fig 6A). $K_i$ values for adenine inhibition of 6-MP uptake were 25 ± 9 μM and 51 ± 13 μM for HEK293-hSLC43A3 and HEK293-mslc43a3 cells, respectively. These $K_i$ values are similar to the $K_m$ values determined for [³H]adenine uptake in the two cell lines. Likewise, 6-MP also exhibited concentration-dependent inhibition of ENBT1-mediated [³H]adenine uptake, but complete inhibition was not obtained at the highest concentration of 6-MP tested (1 mM) (Fig 6B). $K_i$ values for 6-MP inhibition of adenine uptake were 5.9 ± 0.7 mM and 7.4 ± 1.3 mM for HEK293-hSLC43A3 and HEK293-mslc43a3 cells, respectively, which are not significantly

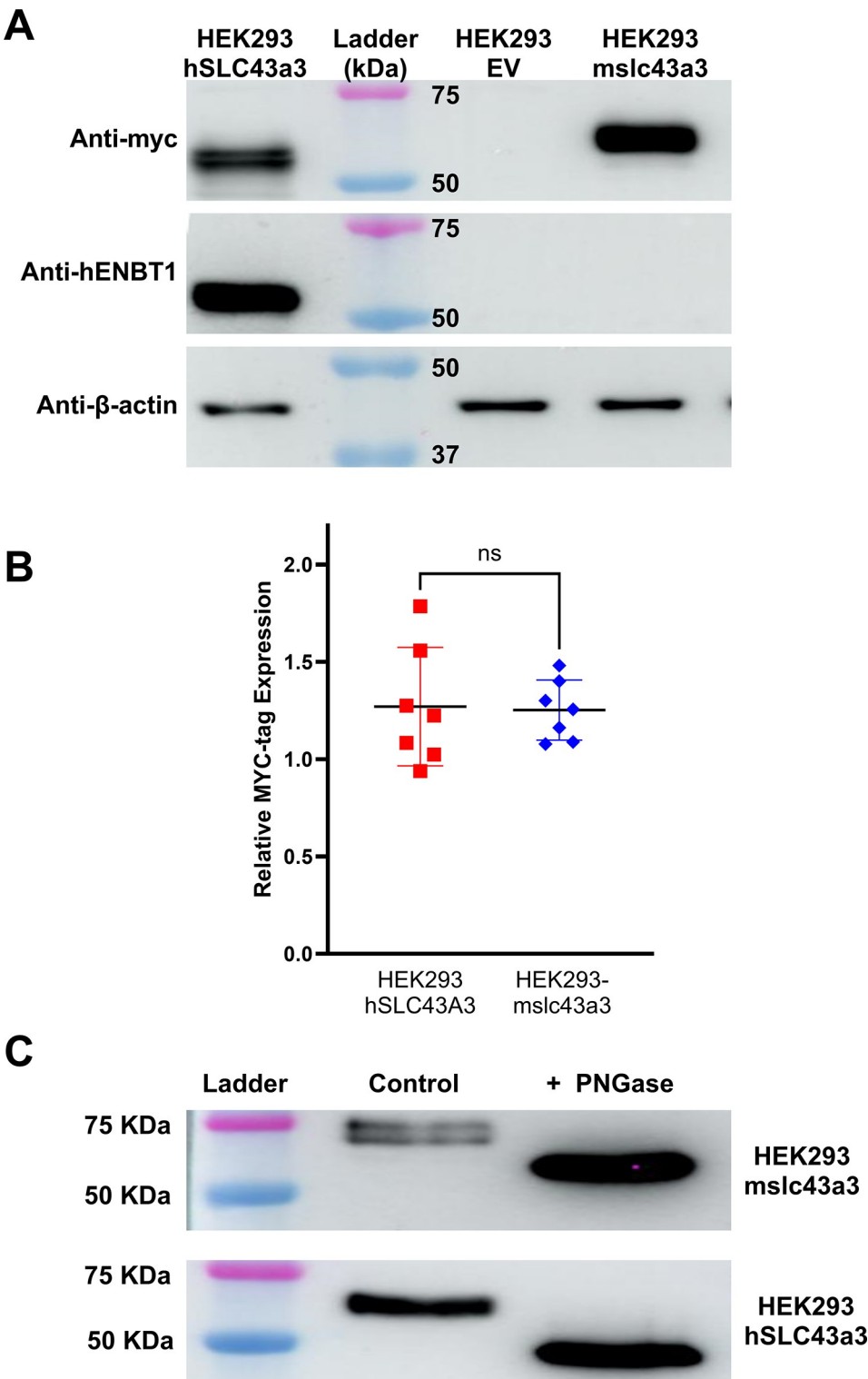

**Fig 4. Immunoblot analysis of ENBT1 expression.** (A) Representative immunoblot showing ENBT1 protein levels in HEK293 cells transfected with pcDNA3.1(-) empty vector (HEK293-EV), or vector constructs containing MYC-tagged *hSLC43A3* (HEK293-hSLC43a3) or MYC-tagged *mslc43a3* (HEK293-mslc43a3). Blots were probed with anti-MYC, anti-hENBT1, or anti-β-actin antibody, as indicated. A protein ladder is shown with the corresponding kDa standards for reference. (B) Quantification of the anti-MYC bands obtained as shown in Panel A relative to the β-actin band from 7 independent cell samples. Mean expression is shown as a horizontal bar and statistical analysis was done using

an unpaired Students t-test ('ns'—no significant difference). (C) To assess the degree of glycosylation of the expressed mENBT1 (top) or hENBT1 (bottom), proteins extracted from the *mslc43a3* and *hSLC43A3*-transfected HEK293 cells were incubated with 500 units of PNGase-F (+PNGase) or in the absence of PNGase-F (Control) at 37°C for 1 h to de-glycosylate proteins, then separated on SDS-PAGE gels and probed with anti-MYC primary antibody as described for Panel A.

different. However, the $K_i$ values estimated for 6-MP inhibition of [$^3$H]adenine uptake were at least an order of magnitude higher than the $K_m$ determined for the uptake of [$^3$H]6-MP by both mENBT1 and hENBT1.

## [$^3$H]6-MP uptake in MOLT-4 and L1210 cells

Time course assays were performed to determine an appropriate estimation of the initial rate of 30 μM [$^3$H]6-MP uptake by endogenous ENBT1 expressed in MOLT-4 and L1210 cells (Fig 7). Both cell lines exhibited rapid time-dependent uptake of [$^3$H]6-MP, plateauing at roughly 30 pmol/μL (similar to the initial extracellular concentration of substrate added, as expected for an equilibrative transport system), with similar k values of $0.34 \pm 0.09$ seconds$^{-1}$ for MOLT-4 cells and $0.22 \pm 0.05$ seconds$^{-1}$ for L1210 cells. The uptake of [$^3$H]6-MP in these cells was significantly inhibited by 5 mM adenine, with no apparent time-dependence to the cellular accumulation in the presence of this concentration of adenine. Based on these results, the lowest achievable time point of 2 seconds was again chosen as the best estimation of initial rate. Similar assays were then performed, using this 2 second incubation time, to determine substrate transport kinetics in MOLT-4 and L1210 cells (Fig 8). Transporter-mediated [$^3$H]

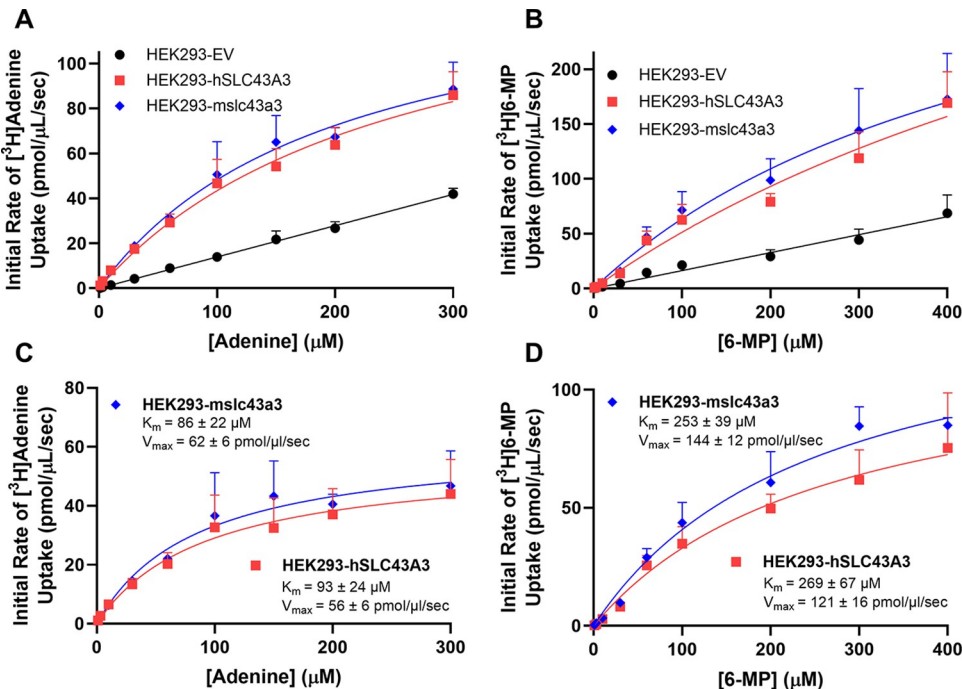

**Fig 5. ENBT1-mediated substrate uptake in transfected-HEK293 cells.** Cells transfected with pcDNA3.1 empty vector (HEK293-EV) or vector constructs containing *hSLC43A3* (HEK293-hSLC43A3) or *mslc43a3* (HEK293-mslc43a3) where incubated with a range of concentrations of [$^3$H]adenine (A) or [$^3$H]-6-MP (C) for 2 seconds. Transporter-mediated uptake shown in Panels B and D was calculated as the uptake in the transfected cells lines minus that measured in the HEK293-EV cells. Each point is the mean $\pm$ SD, n = 6.

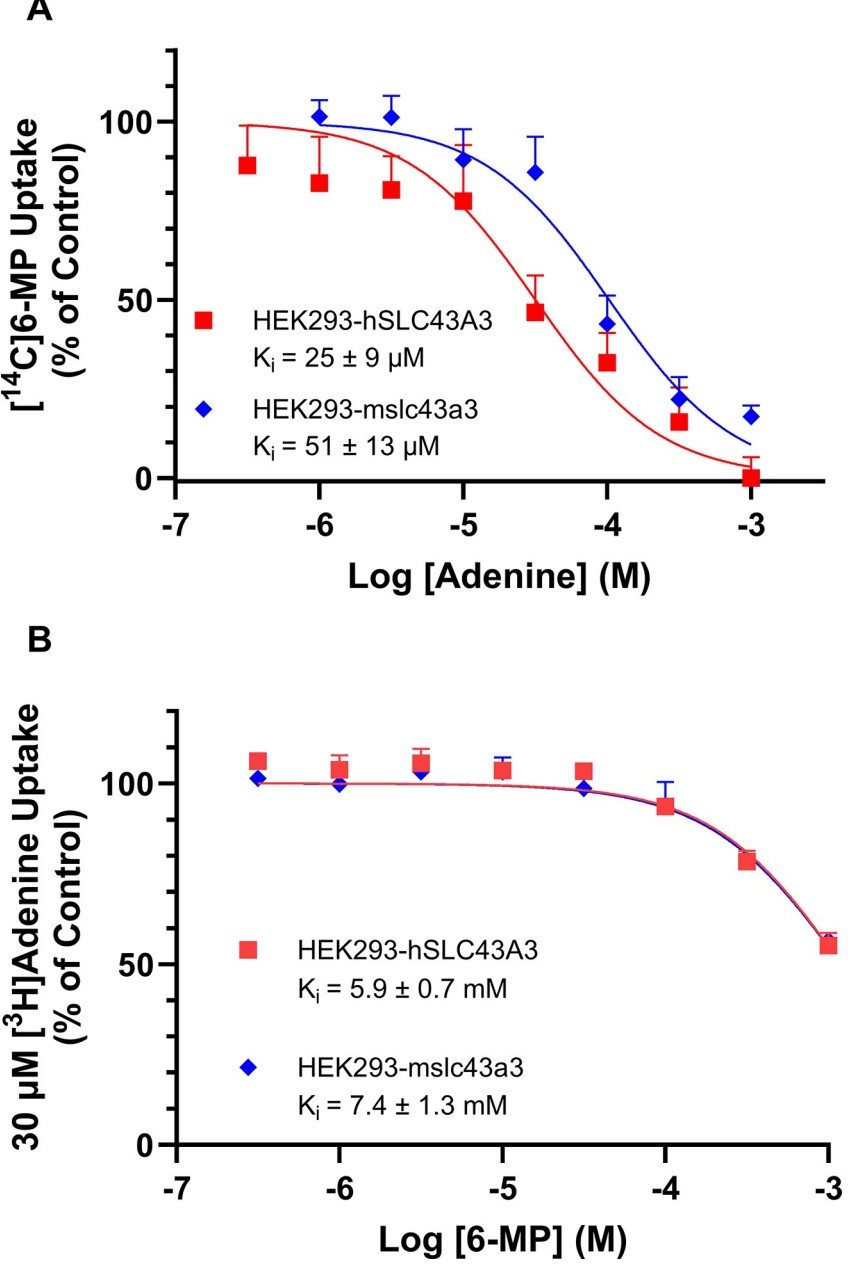

**Fig 6. Reciprocal inhibition of ENBT1-mediated [³H]adenine and [³H]6-MP uptake by 6-MP and adenine.**
HEK293 cells transfected with *hSLC43A3* or *mslc43a3* were incubated with (A) 100 μM or 150 μM [¹⁴C]6-MP, respectively, for 2 seconds in the presence or absence (Control) of a range of concentrations of adenine, or (B) 30 μM [³H]adenine for 2 seconds in the presence or absence (Control) of a range of concentrations of 6-MP. In each case, non-mediated uptake of radiolabeled substrate was determined in the presence of 5 mM adenine and subtracted prior to calculating % of Control uptake as the uptake in the presence of inhibitor divided by the uptake in the absence of inhibitor (X100). Each point represents the mean ± SD, n = 5.

6-MP uptake was found to be saturable in both MOLT-4 and L1210 cells with $V_{max}$ values of 39 ± 9 pmol/μL/second and 81 ± 15 pmol/μL/sec, respectively. These $V_{max}$ values were significantly different between the two cell lines (Extra sum-of-squares F-test, F1,87 = 4.660, p = 0.034, n = 5), reflecting a higher level of ENBT1 expression in the L1210 cells. In contrast,

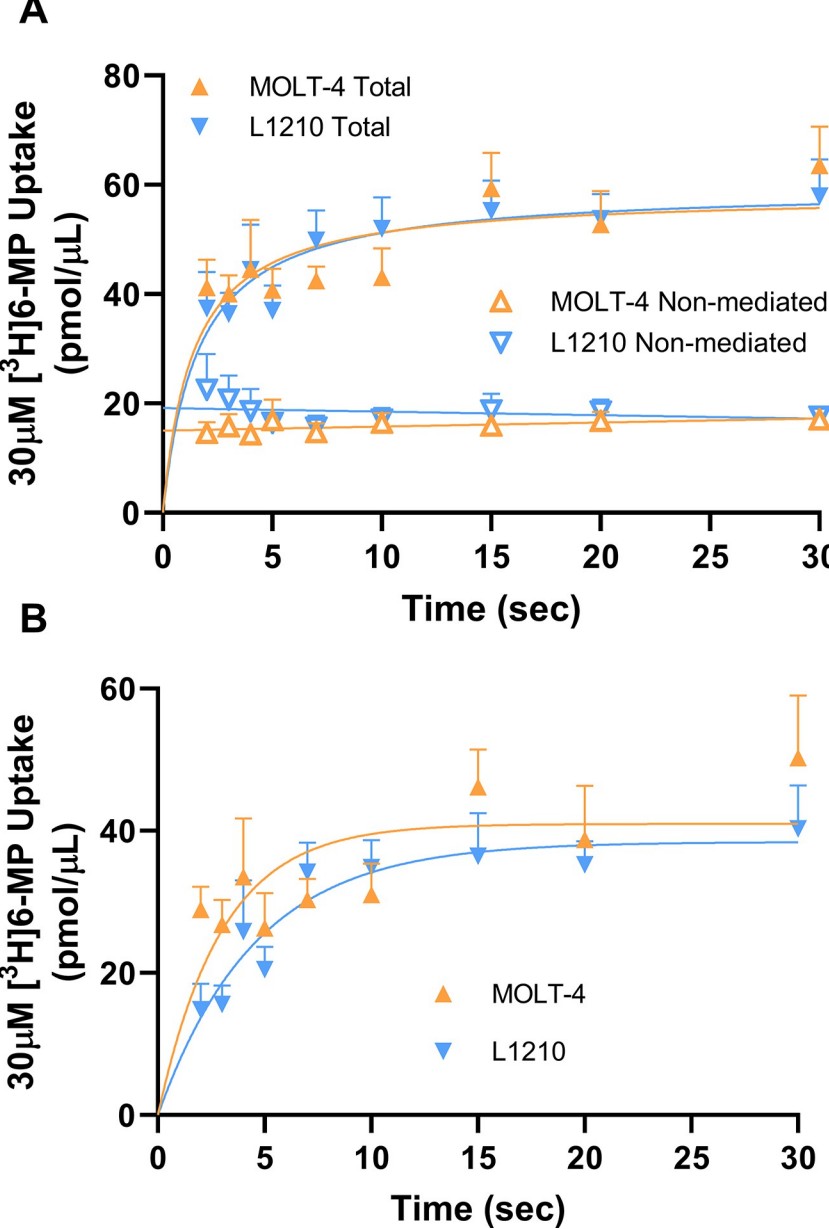

**Fig 7. Time course of [³H]6-MP uptake by MOLT-4 and L1210 cells.** (A) Cells were incubated with 30 μM [³H] 6-MP for the indicated times in the absence (Total uptake) and presence (Non-mediated uptake) of 5 mM adenine. (B) ENBT1-mediated transport of [³H]6-MP calculated as the difference between the Total uptake and Non-mediated uptake shown in Panel A. Each point represents the mean ± SD, n = 5.

The $K_m$ value for the MOLT-4 cells tended to be lower than that for the L1210 cells (Fig 8), but this difference was not found to be statistically significant.

## Effect of 6-MP and its metabolites on cell viability

Transfected HEK293 cells exhibited a biphasic 6-MP cell viability curve with a 19-fold shift in first-phase $EC_{50}$ in the HEK293-hSLC43A3 ($EC_{50}$ = 0.99 ± 0.09 μM) and HEK293-mslc43a3 cells ($EC_{50}$ = 0.96 ± 0.28 μM), compared with that seen in the HEK293-EV cells ($EC_{50}$ =

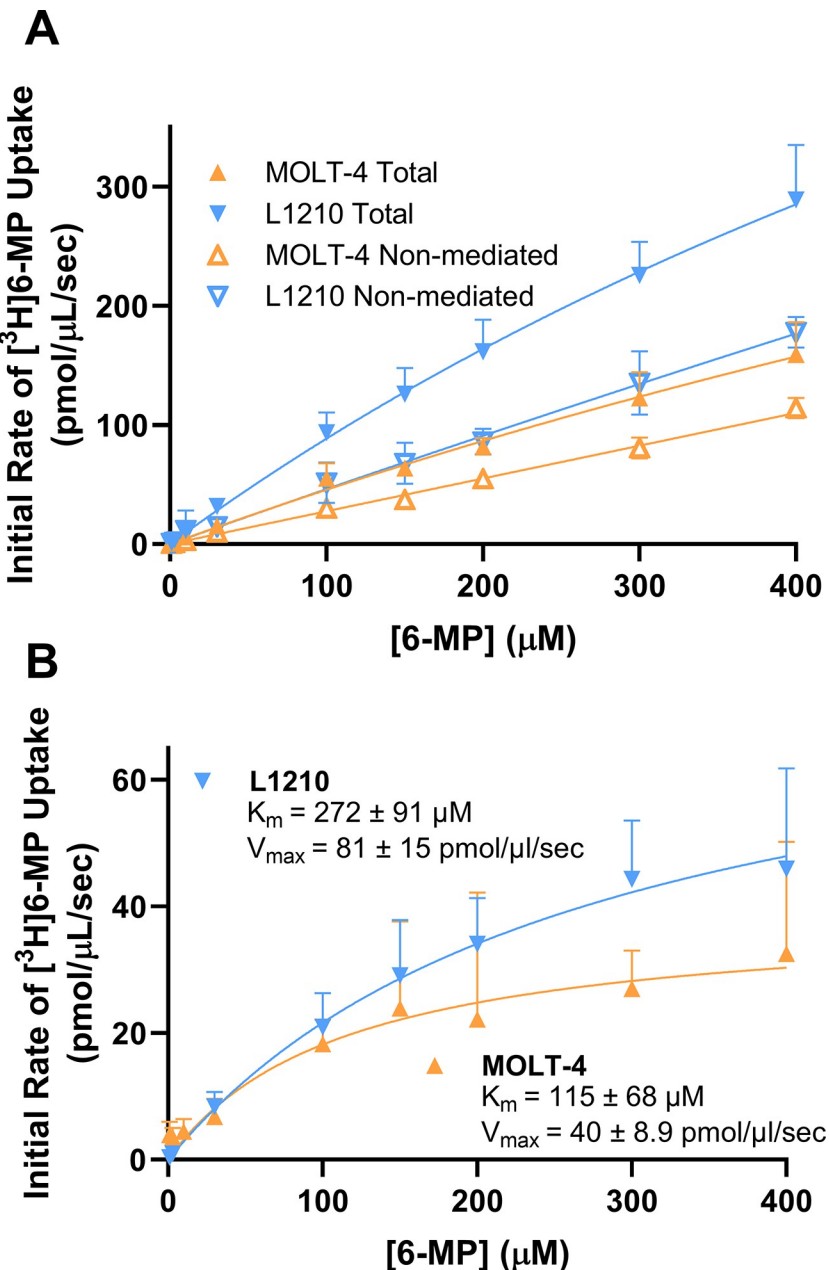

**Fig 8. ENBT1-mediated substrate uptake in MOLT-4 and L1210 cells.** (A) Cells were incubated with a range of concentrations of [$^3$H]6-MP for 2 seconds in the absence (Total uptake) or presence (Non-mediated uptake) of 5 mM adenine. (B) Transporter-mediated uptake of [$^3$H]6-MP by MOLT-4 and L1210 cells calculated as the difference between the Total uptake and the Non-mediated uptake. Each point is the mean ± SD, n = 6.

18.8 ± 0.51 μM) (Fig 9A). In contrast, HEK293 cells transfected with either *hSLC43A3* or *mslc43a3* exhibited a single-phase 6-TG cell viability curve with a 5-fold shift in $EC_{50}$ (HEK293-hSLC43A3 $EC_{50}$ = 3.4 ± 0.9 μM; HEK293-mslc43a3 $EC_{50}$ = 3.4 ± 1.1 μM) relative to the HEK293-EV cells ($EC_{50}$ = 16 ± 4.3 μM) (Fig 9B). Transfected HEK293 cells also exhibited a biphasic 6-MeMP cell viability profile, but unlike that seen with 6-MP, the effect of 6-MeMP was similar in all three cell models (HEK293-EV $EC_{50}$ = 45 ± 7 μM; HEK293-hSLC43A3 $EC_{50}$ = 79 ± 2 μM; HEK293-mslc43a3 $EC_{50}$ = 42 ± 6 μM) (Fig 9C). 6-TU had minimal effect on cell

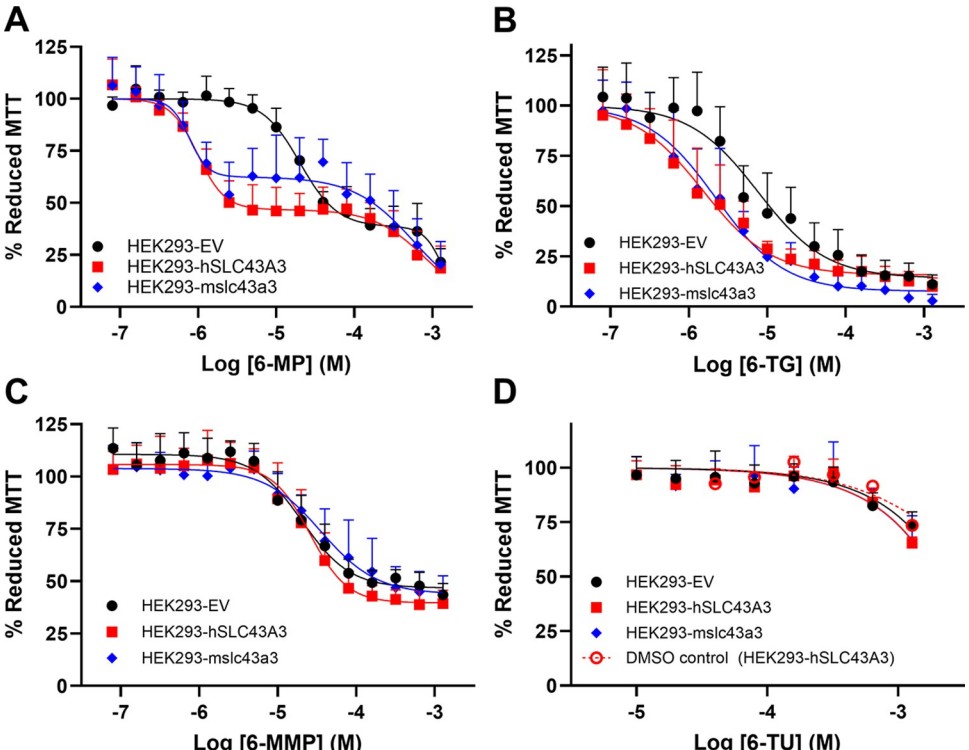

**Fig 9. Effect of 6-MP and its metabolites on HEK293 cell viability in the absence and presence of ENBT1 expression.** HEK293-hSLC43A3, HEK293-mslc43a3, and HEK293-EV cells were treated with a range of concentrations of 6-MP (A), 6-TG (B), 6-MeMP (C) or 6-TU (D) for 48 h at 37°C and cell viability was then assessed using the MTT assay. Panel D also shows the effect of the solvent, DMSO, alone on the viability of HEK293-hSLC43A3 cells. Each point represents the mean ± SD of 9 (A), 7 (B), 11 (C) and 6 (D) experiments.

viability in all of the cell lines tested, with the decrease in cell viability seen at the higher concentrations similar to that produced by the solvent DMSO alone (Fig 9D). MOLT-4 cells exhibited a biphasic 6-MP cell viability curve with first-phase $EC_{50}$ of $0.7 \pm 0.2$ μM. On the other hand, L1210 cells exhibited a single-phase 6-MP cell viability curve with $EC_{50}$ of $0.6 \pm 0.2$ μM (Fig 10A). To evaluate effect of 1 μM DY on 6-MP cell viability, which is present in all transport assays to block ENT1/ENT2 mediated transport, 6-MP cell viability assays were replicated in MOLT-4 cells in the presence or absence of 1 μM DY (Fig 10B). MOLT-4 cells exhibited a biphasic 6-MP cell viability curve with a first-phase $EC_{50}$ of $0.3 \pm 0.1$ μM in presence of DY, and a first-phase $EC_{50}$ of $0.5 \pm 0.1$ μM in absence of DY (but with the solvent, DMSO, present). The 6-MP $EC_{50}$ was not significantly different between MOLT-4 and L1210 cells. However, the $EC_{50}$ for the 6-MP induced decrease in cell viability was significantly different between MOLT-4 cells treated with 6-MP + DY and cells treated with 6-MP + solvent (DMSO) (Extra sum-of-squares F-test, $F_{1, 166} = 6.911$, $p = 0.009$, $n = 6$).

## Discussion

*SLC43A3* encodes for the equilibrative nucleobase transporter ENBT1 [13]. This transporter has been shown to mediate the flux of the nucleobase analogue drug 6-MP across the cell plasma membrane [14]. We have previously shown, via over-expression and shRNAi knockdown studies in human cell models, that the level of expression of this transporter impacts the effect of 6-MP on cell viability [18]. To further pursue the physiological roles of this novel

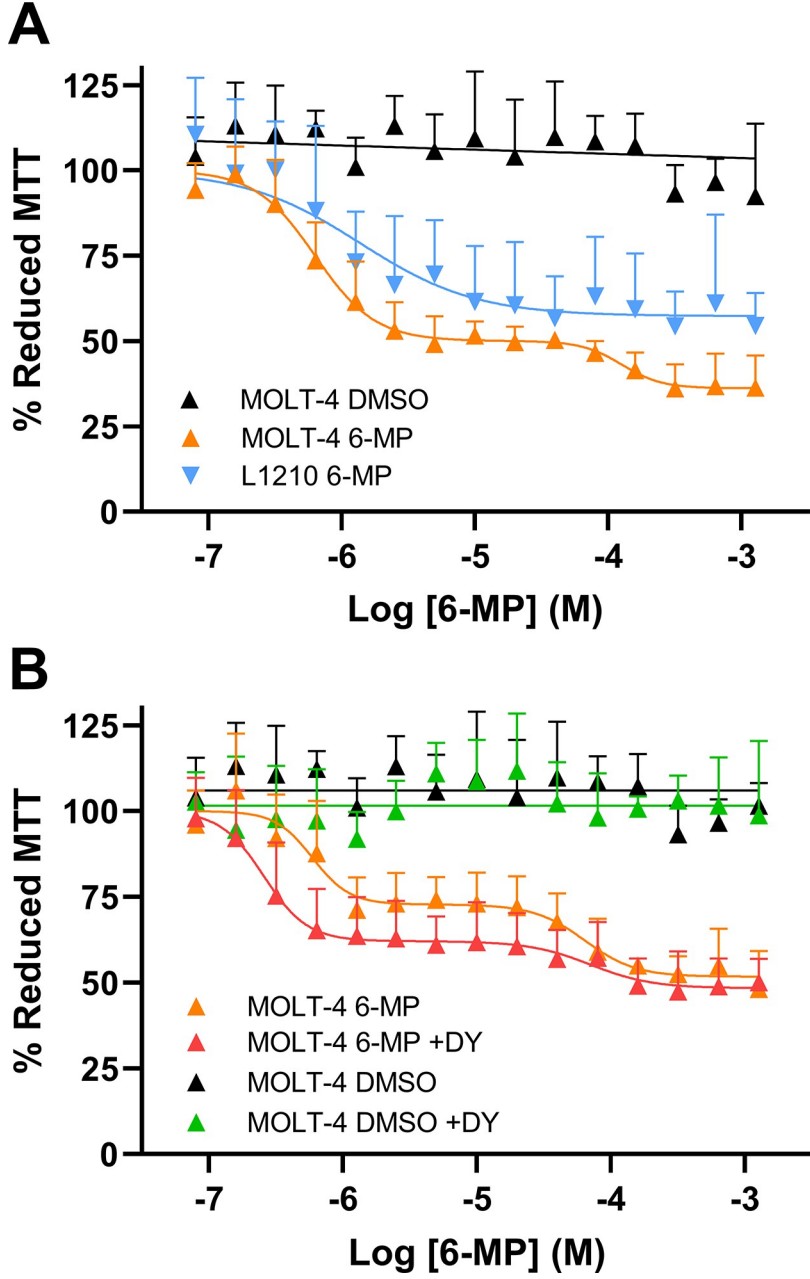

**Fig 10. Effect of 6-MP on MOLT-4 and L1210 cell viability and the effect of dipyridamole (DY) thereon.** (A) Cells were treated with the indicated concentrations of 6-MP or the comparable concentrations of DMSO (solvent) for 48 h at 37°C and cell viability was then assessed using the MTT assay. (B) Cells were treated with the indicated concentrations of 6-MP in the presence (+DY) and absence of 1 μM dipyridamole for 48 h at 37°C and cell viability was then assessed using the MTT assay. Each point represents the mean ± SD of 5 (A) or 6 (B) experiments.

transporter, and assess how its loss would impact 6-MP pharmacokinetics, we plan on developing a *slc43a3* knockout mouse (C57BL/6) model. However, there is currently no information available on how *slc43a3*/ENBT1 expression and function in mice compares with that of humans. The current study remedies that. Overall, our data show that *slc43a3*/ENBT1 expression and function, as well as its ability to transport 6-MP in cells of murine origin is similar to what is seen in human cells. In addition, the tissue expression of *slc43a3* in C57BL/6 mice is

similar to that of *SLC43A3* in humans in that lung and liver express relatively high levels of this transcript and very low levels are found in brain [20]. While transcript expression is similar, the possibility still remains that there are differences in how the plasma membrane expression of mouse and human ENBT1 protein is regulated. Specifically, the notable polyglutamine tract in the central intracellular loop of murine ENBT1 could potentially affect its intracellular trafficking via cytosolic protein interactions [21]. Further examination of this possibility awaits the availability of a specific mouse ENBT1 antibody.

Adenine and 6-MP $K_m$ values obtained in the present study using *SLC43A3*-transfected-HEK293 and MOLT-4 cells were consistent with those published previously by our laboratory group [14, 18]. Notably, *SLC43A3*/hENBT1 and *slc43a3*/mENBT1 $K_m$ values for adenine and 6-MP were similar in all cell lines examined, including those expressing recombinant ENBT1 and endogenously expressing cell lines. This indicates that human and murine variants of ENBT1 transport adenine and 6-MP with similar affinities. However, in each case, the 6-MP $V_{max}$ was 3-fold higher than adenine $V_{max}$. This suggests a difference in how each of the substrates are handled by the transporter. Nevertheless, given that the 6-MP $K_m$ was also 3-fold higher than adenine $K_m$ the translocation efficiencies ($K_m/V_{max}$) of the two substrates were similar. The difference may reflect differing rates of ENBT1-substrate complex decomposition ($K_{off}$) between adenine and 6-MP due to differing binding affinities [22, 23]; adenine has a high affinity for ENBT1 suggesting a slower dissociation rate, thus making the transporter easier to saturate. In an alternating-access model of transporter function, which is common in many solute carriers [24, 25], the substrate has to dissociate from the intracellular-facing conformation before it can return to its empty extracellular-facing conformation. It is possible that the higher affinity substrate leads to a slower 'resetting' of the transporter leading to a reduced apparent $V_{max}$.

$K_i$ values for adenine and 6-MP inhibition of *SLC43A3*/hENBT1 and *slc43a3*/mENBT1 were also similar, suggesting human and murine variants of ENBT1 share similar inhibition kinetics. Under the assumption that in the case of a purely competitive inhibitor, $K_m = K_i$ [19], adenine appears to be a purely competitive ENBT1 inhibitor. However, 6-MP had a $K_i$ that was 30-fold higher than its $K_m$ for the transporter. This may again be due to differing substrate-ENBT1 $K_{off}$ values and warrants further investigation.

The effect of 6-MP and its metabolites on cell viability was assessed under the hypothesis that if ENBT1 mediated the uptake of a cytotoxic compound, or one that is otherwise detrimental to cellular mitochondria function, cells which express hENBT1 and mENBT1 would exhibit higher sensitivity to the compound, relative to cells that do not express either transporter. Concurring with expression and uptake results, HEK293-hSLC43A3 and HEK293-mslc43a3 cells exhibited a 19-fold increase to 6-MP sensitivity relative to HEK293-EV cells. 6-TG, another clinically relevant thiopurine, exhibited 5-fold increased sensitivity in HEK293-hSLC43A3 and HEK293-mslc43a3 cells, relative to HEK293-EV cells. Based on previous findings showing that 6-TG competitively inhibits 6-MP uptake [14] this suggests that 6-TG is also a substrate for ENBT1. It is also noteworthy that, unlike 6-MP, the concentration response profile for the 6-TG effect on cell viability was monophasic. This may reflect the fact that 6-TG enters the intracellular metabolic pathway at a later stage than 6-MP and is converted to the active 6-thioguanine nucleotides directly, leading to a more profound and rapid effect on cell viability. Previous studies have also shown that the 6-MP metabolite 6-MeMP inhibited the uptake of 6-MP by ENBT1 [14]. However, transfection of HEK293 cells with either hENBT1 or mENBT1 had no effect on the ability of 6-MeMP to decrease cell viability, indicating that it is unlikely to be a substrate for ENBT1. To confirm that the similarities in mENBT1 and hENBT1 function seen in the HEK293 over-expression models were not due to an artifact of the expression system, we also examined the characteristics of ENBT1-mediated

[$^3$H]6-MP uptake in a commonly used human leukemia cell model, MOLT-4, and a commonly used mouse leukemia cell model, L1210. Both cell lines had similar initial rates of influx and $K_m$ values for ENBT1-mediated transport of 6-MP, and both cell lines were also similar in their sensitivity to 6-MP with respect to decreasing cell viability. Interestingly, co-treatment of MOLT-4 cells with 6-MP and the equilibrative nucleoside transporter inhibitor DY caused a 2-fold increase in 6-MP sensitivity with respect to its ability to decrease cell viability. This is likely due to DY preventing exogenous nucleoside salvage via the ENTs leading to reduced competition with 6-MP for its metabolic enzymes and subsequent incorporation as a false-nucleotide during DNA synthesis [10].

## Conclusion

This study shows that hENBT1 and mENBT1 are functionally similar in mediating the plasma membrane flux of 6-MP and its effects on cell viability. Therefore, mouse models may be used to obtain clinically relevant data on the role of ENBT1 in the actions of 6-MP. It is possible that variations in the expression of ENBT1, and/or polymorphisms in the *SLC43A3* gene, may underlie some of the observed inter-individual differences in 6-MP pharmacokinetics and the incidence and severity of its adverse effects when used in the treatment of acute lymphoblastic leukemia and intestinal bowel diseases. While 28 polymorphisms of *SLC43A3* have been identified to date [26], there is no information on how these polymorphisms affect transporter function or expression. A *slc43a3* knock-out mouse model may also prove useful for discerning other physiological roles and potential pathophysiological involvement of this novel nucleobase transporter.

## Supporting information

**S1 Raw images.**
(PDF)

## Acknowledgments

The expert technical assistance of Deborah Sosnowski and Tierah Hinchliffe in support of this work is gratefully acknowledged.

## Author Contributions

**Conceptualization:** James R. Hammond.

**Data curation:** James R. Hammond.

**Formal analysis:** Chan S. Kim, Aaron L. Sayler, Hannah Dean, Nicholas M. Ruel.

**Funding acquisition:** James R. Hammond.

**Investigation:** Chan S. Kim, Aaron L. Sayler, Hannah Dean, Nicholas M. Ruel.

**Methodology:** Chan S. Kim.

**Project administration:** James R. Hammond.

**Supervision:** James R. Hammond.

**Writing – original draft:** Chan S. Kim.

**Writing – review & editing:** Chan S. Kim, Nicholas M. Ruel, James R. Hammond.

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
