## [Decision Letter · Decision Letter 0]

13 Aug 2024

PONE-D-24-27708Functional comparison of human and murine equilibrative nucleobase transporter 1PLOS ONE

Dear Dr. Hammond,

Thank you for submitting your manuscript to PLOS ONE. After careful consideration, we feel that it has merit but does not fully meet PLOS ONE’s publication criteria as it currently stands. Therefore, we invite you to submit a revised version of the manuscript that addresses the points raised during the review process. Specifically, please 1) carefully review your figures and figure legends in consideration of the reviewers comments to determine what is necessary and sufficient to clearly communicate the information, and to ensure that they are clearly and well labeled, as needed; 2) ensure that the appropriate work is cited throughout (note that you may choose to include citations suggested by reviewers if appropriate, but this is not a requirement); and 3) as necessary, clarify within the text and figures the points raised by the reviewers that are ambiguous or require further description/analysis.     

Please include the following items when submitting your revised manuscript:A rebuttal letter that responds to each point raised by the academic editor and reviewer(s). You should upload this letter as a separate file labeled 'Response to Reviewers'.A marked-up copy of your manuscript that highlights changes made to the original version. You should upload this as a separate file labeled 'Revised Manuscript with Track Changes'.An unmarked version of your revised paper without tracked changes. You should upload this as a separate file labeled 'Manuscript'.If applicable, we recommend that you deposit your laboratory protocols in protocols.io to enhance the reproducibility of your results. Protocols.io assigns your protocol its own identifier (DOI) so that it can be cited independently in the future. For instructions see: https://journals.plos.org/plosone/s/submission-guidelines#loc-laboratory-protocols. Additionally, PLOS ONE offers an option for publishing peer-reviewed Lab Protocol articles, which describe protocols hosted on protocols.io. Read more information on sharing protocols at https://plos.org/protocols?utm_medium=editorial-email&utm_source=authorletters&utm_campaign=protocols.

We look forward to receiving your revised manuscript.

Kind regards,

Jarrod B. French, PhD

Academic Editor

PLOS ONE

Journal Requirements:

2. Thank you for stating the following financial disclosure: Canadian Institutes of Health Research, Operating Grant#168913

Reviewers' comments:

Reviewer's Responses to Questions

**Comments to the Author**

1. Is the manuscript technically sound, and do the data support the conclusions?

Reviewer #1: Yes

Reviewer #2: Yes

2. Has the statistical analysis been performed appropriately and rigorously? 

Reviewer #1: Yes

Reviewer #2: Yes

3. Have the authors made all data underlying the findings in their manuscript fully available?

Reviewer #1: Yes

Reviewer #2: No

4. Is the manuscript presented in an intelligible fashion and written in standard English?

Reviewer #1: Yes

Reviewer #2: No

5. Review Comments to the Author

Reviewer #1: Summary – This manuscript from Dr. Hammond’s laboratory is focused on a well-studied polytopic integral membrane protein involved in mediating nucleobase transport across cell membranes. This protein, equilibrative nucleobase transporter 1 or ENBT1, has been previously studied by the Hammond lab with the current manuscript focused on comparing a murine model with in vitro human cell lines. The authors specifically assess differences in transporter expression and impact on drug response and toxicity. Presented studies demonstrate that murine (mENBT1) and human (hENBT1) exhibit similar transport kinetics and substrate specificities for 6-mercaptopurine (6-MP) and adenine. Authors argue that these findings validate use of murine models for studying 6-MP pharmacokinetics and adverse effects. Additionally, results suggest that both human and mouse orthologs impact 6-MP-induced cytotoxicity, suggesting that variations in ENBT1 expression could influence therapeutic outcomes and adverse effects in clinical settings. These findings support further exploration of ENBT1’s physiological roles and its potential as a therapeutic target – inclusive of the mouse model presented in this manuscript. This is essential work towards leveraging a knockout murine model under development. Findings presented are of merit for publication though the manuscript, as reviewed during this submission, requires substantial revision prior to publication. These are outlined below.

General comments: Seems some key references are missing from this submission, please review submission for opportunities to include additional relevant citations and ensure correct format. One example is the 2004 paper from Baldwin, Cass, and others on SLC29s that helps support linkage of transporter expression levels on drug efficacy and toxicity. (Note: This reviewer is not affiliated with the above paper). Narrative flow can be improved by breaking down longer sentences and paragraphs. Ensure consistent use of abbreviations and define them upon first use. The Discussion section is well written overall yet raises several key questions about 6-MP and adenine interaction with, and transport by, ENBT1 from both human and murine systems. It seems addressing these questions is an objective of follow-on publications leveraging the model presented. For model validation, the current work is sufficient to support more complete studies.

Line 7 -8: missing punctuation after myelotoxicity

Line 17 – 19: vague on details, what is meant by “in silico” in this instance? Is this just a pairwise sequence alignment or more?

Also, the final abstract statement seems a bit broad in terms of “models” and “actions, adverse effects, and pharmacokinetics”. Is the intent to specifically study ENBT1-mediated action or broader impact beyond specific interactions of the transporter with substrates? The current manuscript doesn’t support mouse models as a general tool for studying disposition and impact of 6-MP relative to human pathology.

Line 24: "various cancers including acute lymphoblastic leukemia (ALL) and autoimmune conditions such like inflammatory bowels disease (IBD) and ulcerative colitis [1]." should be "various cancers including acute lymphoblastic leukemia (ALL) and autoimmune conditions such as inflammatory bowel disease (IBD) and ulcerative colitis [1]."

Line 27: "minimization off-target toxicities" should be "minimization of off-target toxicities."

Line 40: "our lab further characterized ENBT1 in stably transfected human embryonic kidney cells 293 (HEK293) cells" should be "our lab further characterized ENBT1 in stably transfected human embryonic kidney 293 (HEK293) cells."

Line 58: extra space needed “6 -7” vs. “6 – 7”

M&M: Overall, the Materials and Methods are well written with specific detail to reproduce experiments. Well done.

Line 99: Suggest adding an approved IACUC protocol number for the associated animal studies with this statement. Also, "using protocols approved by the Animal Care Committee of the Faculty of Medicine & Dentistry University of Alberta" should be "using protocols approved by the Animal Care Committee of the Faculty of Medicine & Dentistry, University of Alberta."

Line 110: More specific detail on cell culture protocol would be beneficial including concentrations and durations where needed.

Line 225 – 228: The statement "The highest expression levels of slc43a3 were seen in the lung followed by heart liver kidney and spleen with relatively lower expression in duodenum and jejunum." would benefit from quantitative data or a reference figure/table. This statement refers to Figure 3 yet this is just a bar histogram with SEM. Are differences in expression significant between tissue types? Can specific values +/- SEM be added here in parentheses in support of the figure?

Line 410 – 412: “Overall SLC43A3/hENBT1 and slc43a3/mENBT1 had similar effects on 6-MP and 6-TG induced decreases in cell viability further suggesting the human and murine variants of ENBT1 are functionally similar." – Can this be expanded to denote potential clinical significance or broader impacts? Last line of the abstract focuses on “actions, adverse effects, and pharmacokinetics” – there is value in connecting these two at this point.

“Conclusion”: As written, this conclusion seems like a missed opportunity to expand. Either the broader impacts language in the abstract and parts of manuscript should be dialed back or this section can be developed a bit more. This reviewer supports the latter.

Figures: Figures need additional work and polishing.

Figure 1 is cumbersome to understand meaning or intent without work. Other publications in this field have used structures to convey similar changes and pathways where one can more easily see conversions. As shown, this version is difficult to follow and many of these conversions are not relevant to the manuscript as written or discussed.

Figure 2 is full page yet still impossible to define detail noted in the legend. Amino acids are identified within circles but unreadable. Figure is blurry even on screen within the PDF. It’s impossible to see or understand the red markings as noted in the figure legend. It’s important to include this figure – that is clear. But, the current form adds minimal value. Consider changing this graphic to more clearly show differences with larger typeface. Another option is, since the topology is almost identical, just show one panel with topology and a sequence alignment below noting differences in sequences.

Figure 3 would benefit from noting significance between tissues – particularly lung relative to heart – spleen.

Figure 6 – What specifically does “from 6 independent experiments conducted in technical duplicate” mean for transport assays? The mechanics of doing replicates isn’t defined in the transport assay or statistical analysis sections.

General comment on fits – it would be beneficial to include fit values (e.g., Ki or Vmax) within graphs for easy visual comparison. As written, this information is within the main text (results) but not figure legends. One suggestion is to also add these values directly within figures as well.

Reviewer #2: The objective of the studies in this manuscript was to establish that the mouse equilibrative nucleobase transporter 1 or mENTB1 (Slc43a3) is similar in its tissue distribution and function towards 6-mercaptopurine (6-MP) as the human counterpart hENTB1 (SLC43A3). These studies are intended to validate the utility of a mouse null mutant model (Slc43a3-) for mENTB1 that will be used to follow 6-MP pharmcokinetics. Both hENTB1 and mENTB1 share a similar topology and share 73.4% identity at the primary sequence level. Using quantitative RT-PCR analyses on several different tissue types isolated from C57BL/6 mice the authors demonstrate that the tissue distribution for Slc43a3 expression is similar to that observed for SLC43A3 in human tissue. Kinetic analyses on HEK293 cells expressing the coding sequences SLC43A3 (hENTB1) and Slc43a3 (mENTB1) indicated that the kinetic properties of these transporters are highly similar with respect to the ligands, adenine and 6-mercaptopurine. Overall, the studies in the manuscript are well executed and thorough. However, there are a few minor points that should be addressed to improve the overall clarity of the manuscript.

1. Consider removing Figure 1. I’m not sure that it is relevant or adds much to the study. If the figure is still included, it might be useful to indicate which activities are primarily affected by genetic variations.

2. Is anything known about genetic variations in SLC43A3 and 6-MP toxicity? The authors allude to a role for this transporter in 6-MP bioavailability in the introduction, but I’m confused whether genetic variations in this transporter have been described. Please clarify this point in the introduction.

3. The presence of the polyglutamine tract in mENTB1 (slc43a3) is interesting and potentially could provide a site of interaction between mENTB1 transporters, or with other cytosolic proteins. What is known about the quaternary structure for mENTB1 and is it different than that of hENTB1? Perhaps the authors could provide some additional discussion about the significance of the polyglutamine tract in the manuscript.

4. Figure 3 and Ln 227: I suggest changing the order in the text of the tissues to match the order in the figure (heart, kidney, liver, and spleen). Were the differences in the relative expression levels statistically significant? Please include that information in the figure or text. Additionally, please specify that the tissue distribution is for the mouse strain C57BL/6 in the text. Is this the same mouse strain used to create the null mutant? If so, please include in the discussion.

5. Ln 237 - 239: In text mention that qualitative PCR and immunoblot were performed using cDNA and lysates from HEK293 transfected cells, as well as the MOLT-4 and L1210 cell lines and this data is shown in figure 4. However, the data in Figure 4 shows just the PCR results for the HEK293 transfected cell lines. Additionally, please indicate the primer pairs used for the PCR analysis, as well as the PCR conditions, in the figure legend. If these are the same as for the qPCR, please indicate as such.

6. Ln 241 -243: The data or citation for the expression of SLC43A3 in MOLT-4 cells and Slc43a3 in L1210 cells doesn’t appear to have been included?

7. Lns 273-275: Consider moving the Km and Vmax values (as well as the Ki values) to a table format for conciseness and ease of comparison.

8. Figure 7B and Lns 290-295: I’m intrigued by the inhibition data using 6-MP as an inhibitor of adenine uptake. The data suggest that the inhibition mechanism might be more complex than just competitive inhibition. I would suggest that the authors provide a more detailed discussion for the underlying reasons for this observation and expand on the comment in the discussion that it may be due to differing substrate-ENTB1 Koff values.

9. Figure 8: The data points for 6-MP uptake in MOLT-4 and L1210 cells show substantial deviation and are not particularly convincing of a linear correlation between 6-MP uptake with time. Indeed, in the MOLT-4 cells the difference in 6-MP uptake doesn’t seem to change between 2 sec and 20 sec? Does this reflect solely binding of 6-MP to the transporter and not uptake per se? I’m curious if the authors have tried extending the time course (to say 5 or 10 mins) and what those results reflected?

10. Figures 8A and 9A: Consider changing the symbols in these figures to open and closed symbols to more clearly differentiate between the total and non-mediated ENTB1 uptake. When the symbols overlay it is difficult to differentiate between symbols of the same color and shape.

11. Figures 10 and 11: The authors speculate that the difference between the biphasic and monophasic EC50 curves for TG and 6-MP reflects their different routes of metabolism. I’m a little confused by that interpretation. Wouldn’t an EC50 values in the same population be monophasic regardless of which intracellular targets the drug interacts with? I would suggest instead that this reflects different subpopulations within the experiment, which seems quite likely for Figure 10 when dealing with transfected cells.

6. PLOS authors have the option to publish the peer review history of their article (what does this mean?). If published, this will include your full peer review and any attached files.

Reviewer #1: No

Reviewer #2: No

---

## [Author Response · Author response to Decision Letter 0]

11 Sep 2024

See "Response to reviewers" document submitted with this resubmission.

---

## [Editor Report · Decision Letter 1]

20 Sep 2024

Functional comparison of human and murine equilibrative nucleobase transporter 1

PONE-D-24-27708R1

Dear Dr. Hammond,

We’re pleased to inform you that your manuscript has been judged scientifically suitable for publication and will be formally accepted for publication once it meets all outstanding technical requirements.

Kind regards,

Jarrod B. French, PhD

Academic Editor

PLOS ONE

---

## [Editor Report · Acceptance letter]

25 Sep 2024

PONE-D-24-27708R1 

PLOS ONE

Dear Dr. Hammond, 

I'm pleased to inform you that your manuscript has been deemed suitable for publication in PLOS ONE. Congratulations! Your manuscript is now being handed over to our production team.

Kind regards, 

on behalf of

Professor Jarrod B. French 

Academic Editor

PLOS ONE